# Chemical Composition and Insecticidal Activities of Essential Oils against the Pulse Beetle

**DOI:** 10.3390/molecules27020568

**Published:** 2022-01-17

**Authors:** C. S. Jayaram, Nandita Chauhan, Shudh Kirti Dolma, S. G. Eswara Reddy

**Affiliations:** 1Entomology Laboratory, Agrotechnology Division, CSIR-Institute of Himalayan Bioresource Technology, Palampur 176061, India; csjayaram182@gmail.com (C.S.J.); nanditachauhan796@gmail.com (N.C.); skdolma@gmail.com (S.K.D.); 2Academy of Scientific and Innovative Research (AcSIR), Ghaziabad 201002, India

**Keywords:** botanicals, essential oils, pulse beetle, fumigant toxicity, synergistic, repellent, ovicidal

## Abstract

Pulse beetles, *Callosobruchus chinensis* and *Callosobruchus maculatus*, are essential pests of cowpea, gram, soybean and pulses. Application of synthetic insecticides against the pulse beetle has led to insect resistance; insecticide residues on grains affect human health and the environment. Essential oils (EOs) are the best alternatives to synthetics due to their safety to the environment and health. The main objective of the investigation was to study the chemical composition and insecticidal activities of EOs, their combinations and compounds against the pulse beetle under laboratory. Neo-isomenthol, carvone and *β*-ocimene are the significant components of tested oils using GC-MS. *Mentha spicata* showed promising fumigant toxicity against *C. chinensis* (LC_50_ = 0.94 µL/mL) and was followed by *M. piperita* (LC_50_ = 0.98 µL/mL), whereas *M. piperita* (LC_50_ = 0.92 µL/mL) against *C. maculatus.* A combination of *Tagetes minuta* + *M. piperita* showed more toxicity against *C. chinensis* after 48 h (LC_50_ = 0.87 µL/mL) than *T. minuta* + *M. spicata* (LC_50_ = 1.07 µL/mL). L-Carvone showed fumigant toxicity against *C. chinensis* after 48 h (LC_50_ = 1.19 µL/mL). Binary mixtures of *T. minuta* +*M. piperita* and *M. spicata* showed promising toxicity and synergistic activity. EOs also exhibited repellence and ovipositional inhibition. The application of *M. piperita* can be recommended for the control of the pulse beetle.

## 1. Introduction

The pulse beetle, *Callosobruchus chinensis* Linnaeus and *Callosobruchus maculatus* Fabricius (Coleoptera: Bruchidae) are the most important pests of food grains and cause damage to cowpea, gram and soybean [1,2,3,4]. The larvae of the pulse beetle feed and develop on the seed, while the adults do not require food and live for 7–14 days. The grubs bore into the grain feed’s internal contents, affecting nutritional quality [2]. In severe infestations, seeds became completely hollow and unsuitable for marketing [5]. The pulse beetle causes more than 50% loss of grains in storage after three to four months [6]. The control of stored grain pests generally depends on synthetic insecticides, including fumigants [2,7]. However, regular and repeated use of insecticides to control stored grain pests has led to insecticide resistance, environmental pollution and human health problems. To reduce complete dependence on synthetic pesticides to control pests on stored grains, the investigation of essential oils (EOs) becomes the option in the current scenario. Organic growers and safety have increased the use of EOs to the environment and consumers [8,9]. Therefore, screening of EOs against the pulse beetle requires an hour for identifying lead(s).

The EOs are extracted from leaves, flowers and bark of aromatic, medicinal, ornamental and other plants, but only 10% of the EOs are used for aroma, flavor and fragrance [10]. EOs will be considered alternatives to conventional chemical pesticides due to their broad spectrum of insecticidal properties, eco-friendliness and potential for commercial use against stored grain pests [8]. EOs contains a complex mixture of phytochemicals (monoterpenes, sesquiterpenes, etc.) exhibiting insecticidal activities against stored grain pests [11,12]. The EO-based formulations are not commercially available to control stored grain pests. Based on this background, the main aims of the present work are to evaluate the selected EOs and their binary mixtures/combinations and the compound for their fumigant, synergistic, repellent and ovicidal activities against the adults of *C. chinensis* and *C. maculatus* for identification of lead(s) for development of botanical formulation.

## 2. Results

### 2.1. Chemical Composition of Essential Oils

The chemical composition of EOs of *Mentha piperita*, *Mentha spicata* and *Tagetes minuta* is present in Appendix A. Results showed about 39 components in three EOs and 4 remained unidentified, accounting for 99.48–100% of the total oils. About 23 components from *M. piperita* and *M. spicata*; and 10 components from *T. minuta* were identified. *neo*-Isomenthol (38.64%) was the main component present in the *M. piperita* followed by menthone (29.54%), *neo*-menthyl acetate (7.55%), menthofuran (6.49%) and 1,8-cineole (6.31%). EO from *M. spicata* had carvone (63.38%) as the primary component followed by limonene (21.30%) and 1,8-cineole (2.29%). Similarly, the EO from *T. minuta* had *β*-ocimene (40.57%) as the primary component followed by dihydrotagetone (28.74%), *Z*-tagetone (11.63%) and *E*-ocimenone (8.72%).

### 2.2. Fumigant Toxicity of EOs and Their Combinations/Binary Mixtures against the Pulse Beetle

The fumigant toxicity results of EOs alone and their combinations, co-toxicity co-efficient and interaction type (synergistic/independent) against *C. chinensis* and *C. maculatus* are present in Table 1 and Table 2.

#### 2.2.1. *Callosobruchus chinensis*

All the tested EOs reported fumigant toxicity against *C. chinensis* (Table 1). Among them, *M. spicata* was found more toxic against adults of *C. chinensis* followed by *M. piperita* and *T. minuta*. The *C. chinensis* was more susceptible to *M. spicata* at 24 and 48 h after treatment (LC_50_ = 1.88 and 0.94 µL/mL), respectively, as compared to *M. piperita* (LC_50_ = 2.06 and 0.98 µL/mL) and *T. minuta* (LC_50_ = 3.49 and 1.41 µL/mL). All the tested EOs are superior to the positive control, i.e., aluminum phosphide (LC_50_ = 0.93 µg/mL) after 72 h. The combination of *T. minuta* oil with *M. piperita* and *M. spicata* showed promising toxicity and synergistic activity against *C. chinensis*. Results showed that (Table 1), *T. minuta* + *M. piperita* (1:1 ratio) showed more toxicity against *C. chinensis* after 24 and 48 h of treatment (LC_50_ = 1.35 and 0.87 µL/mL, respectively) compared to *T. minuta* + *M. spicata* (1.53 and 1.07 µL/mL). However, the combination of EOs was more superior than individual oils of *T. minuta*, *M. piperita* and *M. spicata* at 24 h (LC_50_ = 3.49, 2.06 and 1.88 µL/mL) and 48 h (LC_50_ = 1.41, 0.98 and 0.94 µL/mL), respectively, after treatment.

#### 2.2.2. *Callosobruchus maculatus*

Results from Table 2 indicate that tested EOs showed toxicity against adults of *C. maculatus.* Among them, *C. maculatus* was more susceptible to *M. piperita* after 24 h of treatment (LC_50_ = 1.76 µL/mL) than *T. minuta* at 24 and 48 h (LC_50_ = 3.42 and 2.02 µL/mL), respectively. However, *M. spicata* is inferior to *M. piperita* after 24 h (LC_50_ = 2.74 µL/mL) and superior after 48 h (LC_50_ = 0.92 µL/mL) as compared to *M. piperita* (LC_50_ = 1.12 µL/mL). A combination of *T. minuta* with *M. piperita* and *M. spicata* showed higher toxicity and synergistic activity (Table 2). Similarly, *T. minuta* + *M. spicata* also showed more toxicity and synergistic activity against *C. maculatus* (LC_50_ = 2.42 and 1.42 µL/mL) after 24 and 48 h of treatment, respectively, compared to *T. minuta* + *M. piperita* after 24 h (LC_50_ = 4.93 and 2.40 µL/mL). A combination of *T. minuta* + *M. piperita* did not show synergistic activity. The blends of EOs were not superior to individual oils alone.

### 2.3. Fumigant Toxicity of L-Carvone against C. chinensis and C. maculatus

The results on fumigant toxicity of L-Carvone against *C. chinensis* and *C. maculatus* are presented in Table 3. L-Carvone exhibited promising toxicity against *C. chinensis* after treatment, i.e., 24, 26, 28, 36 and 48 h (LC_50_ = 3.61, 2.92, 2.16, 1.33 and 1.19 µL/mL) as compared to *C. maculatus* (LC_50_ = 6.72, 5.90, 5.39, 3.76 and 3.56 µL/mL).

### 2.4. Repellent Activity of EOs against C. chinensis and C. maculatus

#### 2.4.1. *Callosobruchus chinensis*

The results of repellent activity and repellent indices of *M. piperita*, *M. spicata* and *T. minuta* against *C. chinensis* 1 to 5 h after treatment are present in Appendix A, Table 4 and Table 5 and Figure 1. All the EOs showed repellent activity against *C. chinensis*, and their activity increased as the concentration increased (Figure 1 and Appendix A). In *M. piperita*, a higher concentration (8 µL/mL) showed significantly (F_4,24_ = 8.50–12.79; *p* < 0.0001) higher repellence (88–92%) at 1–4 h intervals after treatment and was followed by 6 µL/mL (68–72%) and 4 µL/mL (24–44%) as compared to other concentrations. Similarly, a higher concentration of *M. spicata* (8 µL/mL) showed significantly (F_4,24_ = 7.73–16.03; *p* < 0.0001) higher repellence (76–84%) at 1–4 h after treatment and was followed by 6 µL/mL (60–72%) and 4 µL/mL (40–44%) than other concentrations. Similarly, *T. minuta* (8 µL/mL) also showed significantly (F_4,24_ = 3.29–23.03; *p* < 0.001–0.03) higher repellence (76–96%) at 1–4 h after treatment and was followed by 6 µL/mL (60–64%) and 4 µL/mL (36–56 %) as compared to other concentrations.

Based on repellent indices (RI), EOs showed repellence (R) against *C. chinensis* at higher concentrations at different hours (h) after treatment (Table 4). In *T. minuta*, all the concentrations (1–8 µL/mL) at 1–4 h after treatment showed repellence (R) against *C. chinensis* except at 1 µL/mL after 4 h showed indifference (I). Similarly, *M. piperita* (2–8 µL/mL) showed repellence (R) after 1–3 h of treatment except 1–4 µL/mL after 4 h and 1 µL/mL after 1 and 2 h showed indifference (I). In the case of *M. spicata*, all the concentrations showed repellence (R) against *C. chinensis* at different hours after treatment except at 1 µL/mL, which showed indifference after 3 and 4 h.

#### 2.4.2. *Callosobruchus maculatus*

The repellent activity of *M. piperita*, *M. spicata* and *T. minuta* oil against *C. maculatus* at 1 to 4 h after treatment is present in Appendix A, Table 5 and Figure 2. All the EOs showed repellent activity against *C. maculatus*. *M. spicata* at 8 µL/mL showed significantly (F_4,24_ = 8.21–12.67; *p* < 0.0001) higher repellence (48–76% repellence) at 1–4 h after treatment and was followed by 6 µL/mL (24–56%) after 1–2 h (F_4,24_ = 8.21–12.67; *p* < 0.0001) and 4 µL/mL (12–40%) as compared to other concentrations. Similarly, the *T. minuta* showed significant differences among the treatments except for 3 h after treatment. *T. minuta* at 8 µL/mL showed significantly higher repellence (51–76%) at different intervals (1, 2 and 3 h) after treatment and was followed by 6 µL/mL (40–52%) and 4 µL/mL (24–40%) as compared to other concentrations. *M. piperita* at 8 µL/mL showed significantly (F_4,24_ = 8.3–9.19; *p* < 0.0001) higher repellence (72–80%) at 1–4 h after treatment and was followed by 6 µL/mL (56–64%) and 4 µL/mL (40–64%) as compared to lower concentrations.

Based on repellent indices (RI), all the tested EOs showed repellence (R) against *C. maculatus* at higher concentrations in different hours (1–4 h) after treatment as compared to other concentrations (Table 5). *T. minuta* (4–8 µL/mL) at different hours (1–4 h) after treatment showed repellence (R) against *C. maculatus* except for 4 µL/mL after 3 h showed indifference (I). However, lower concentrations at 2 µL/mL after 3 and 4 h showed repellence except for 1 µL/mL after 1–4 h showed indifference. Similarly, *M. piperita* (2–8 µL/mL) showed repellence after 1–4 h of treatment except lower concentration (1 µL/mL) which showed indifference after 1–4 h. In *M. spicata*, higher concentrations (2–8 µL/mL) showed repellence against *C. maculatus* at different hours after treatment except for 2 µL/mL after 2 and 4 h and 1 µL/mL after 1–4 h showed indifference.

### 2.5. Oviposition Deterrent Activity of EOs

#### 2.5.1. *Callosobruchus chinensis*

The ovipositional deterrent activity of EOs against *C. chinensis* was present in the Appendix A, Figure 3 and Figure 4. All the tested EOs inhibited the oviposition of *C. chinensis*. *M. spicata* significantly affected the oviposition of *C. chinensis* by inhibiting the oviposition by 100% at 10 µL/mL (F_4,24_ = 53.66; *p* < 0.0001) and was at par with 8 µL/mL after 24 h followed by 95.4 and 92% inhibition after 48 and 72 h, respectively, as compared to other concentrations (Figure 3). Similarly, *T. minuta* at 10 and 8 µL/mL also showed 100% oviposition inhibition after 24 h (F_4,24_ = 15.67; *p* < 0.0001) and 48 h (F_4,24_ = 68.66; *p* < 0.0001). However, *T. minuta* at 10 µL/mL also showed 99.4% inhibition after 72 h (F_4,24_ = 178.98; *p* < 0.0001). *M. piperita* significantly affected the oviposition by inhibiting 100% at 20 µL/mL after 24 h (F_4,24_ = 88.5, *p* < 0.0001) and was followed by 96 and 90.8% after 48 and 72 h, respectively, as compared to other concentrations (Figure 4).

#### 2.5.2. *Callosobruchus maculatus*

The ovipositional deterrent activity of EOs against *C. maculatus* was present in the Appendix A and Figure 5. *M. piperita* significantly affected the oviposition by inhibiting 100% at 12 µL/mL after 24 h (F_4,24_ = 89.86, *p* < 0.0001) and was followed by 94.66% (F_4,24_ = 72.46; *p* < 0.0001) and 77.6% (F_4,24_ = 33.26; *p* < 0.0001) after 48 and 72 h, respectively, as compared to other concentrations. *M. spicata* also significantly affected the oviposition by inhibiting by 94.26% at 12 µL/mL after 24 h (F_4,24_ = 47.75, *P* < 0.0001) and was followed by 91.06 (F_4,24_ = 80.04; *p* < 0.0001) and 75.12%, respectively, after 48 and 72 h as compared to other concentrations. *T. minuta* at 12 µL/mL showed significantly affected the inhibition by 37.24% (F_4,24_ = 4.75, *p* < 0.007) and 35.68% (F_4,24_ = 4.96, *p* < 0.006), respectively, after 48 and 72 h.

### 2.6. Comparative Toxicity, Synergistic, Repellence and Ovipositional Deterrence Activities of EOs and Its Combination against C. chinensis and C. maculatus

Results showed that EOs of *M. piperita, M. spicata* and *T. minuta* showed promising fumigant toxicity against *C. chinensis* (LC_50_ = 0.94–1.41 µL/mL) compared to *C. maculatus* (LC_50_ = 0.92–2.02 µL/mL) after 48 h of treatment. L-Carvone showed more fumigant toxicity against *C. chinensis* (LC_50_ = 1.19 µL/mL) as compared *C. maculatus* (LC_50_ = 3.56 µL/mL). Similarly, a combination of *T. minuta* with *M. piperita* and *M. spicata* (1:1 ratio) showed promising fumigant toxicity against *C. chinensis* (LC_50_ = 0.87–1.07 µL/mL) as compared to *C. maculatus* (LC_50_ = 1.42–2.40 µL/mL) after 48 h of treatment. In synergistic activity, all the combinations of EOs showed synergistic activity against *C. chinensis*, whereas *C. maculatus* showed synergistic activity in the combination of *T. minuta* with *M. spicata*. In repellence activity, *M. piperita, M. spicata* and *T. minuta* at a higher concentration (8 µL/mL) showed significantly higher repellency of 88–92%, 76–84% and 76–96%, respectively, against *C. chinensis* as compared to *C. maculatus* (72–80%, 48–76% and 51–76% repellence, respectively). Similarly, in ovipositional deterrence, *M. spicata* (92–100%) and *T. minuta* (99.4–100%) at a higher concentration (10 µL/mL) showed significantly higher ovipositional inhibition against *C. chinensis* as compared to *C. maculatus* in which *M. spicata* and *T. minuta* at 12 µL/mL showed 75.12–95.26% and 35.68–45.96% inhibition, respectively.

## 3. Discussion

The purpose of the present study is to control the pulse beetle using EOs that are safer, environment friendly and alternative to synthetic insecticides [8,13]. In this study, chemical composition, fumigant, synergistic, repellent and ovipositional activities of EOs of *M. piperita*, *M. spicata* and *T. minuta*, their combinations/binary mixtures and pure compound (L–Carvone) against *C. chinensis* and *C. maculatus* were discussed. The application of EOs, their combination, and the pure compound effectively against *C. chinensis* and *C. maculatus* and reduce the oviposition. EOs are complicated natural mixtures and have different constituents at different concentrations, with few constituents exhibiting insecticidal activities [14]. However, earlier studies also reported synergistic activities between EOs, their combinations and constituents [15,16].

In the present study, the chemical composition of *M. piperita*, *M. spicata* and *T. minuta* oils showed that their primary constituents are neo-Isomenthol and menthone in *M. piperita*; carvone and limonene in *M. spicata*; β-ocimene, dihydrotagetone in *T. minuta*. In addition to significant constituents in EOs, other compounds such as menthyl acetate, menthofuran, 1,8-cineole, limonene, *β*-caryophyllene in *M. piperita*; 1,8-cineole, *β*-myrcene and *cis*-dihydrocarvone in *M. spicata* and *Z*-tagetone, *E*-ocimenone in *T. minuta*, which plays a vital role in the insecticidal activities [14,17]. The variation in the chemical composition and the significant compounds of EOs of *M. piperita*, *M. spicata* and *T. minuta* was observed in the present study. The variation might be due to environmental factors (geographical, seasonal and climatic conditions), genetic/hereditary, chemotype and nutrition of the plants [18].

The tested EOs showed promising fumigant, repellent and ovipositional activities against the pulse beetle. Efficacy of these oils against target pests also depends on the extraction method, season, plant part (leaf, flower, root), chemical constituents, dose/concentration, application method, type of insect and their stage [19]. Numerous studies reported that the insecticidal activities of EOs and their significant components effectively controlled the stored grain pests [20,21]. Most of the investigations endorsed insecticidal activities of EOs due to their significant compounds (neo-Isomenthol, menthone, carvone, limonene, *β*-ocimene and dihydrotagetone), which act on the insect’s nervous system by disturbing the functions of GABAergic [22,23] and aminergic systems [24,25] and by inhibiting the acetylcholinesterase [26,27]. The use of EOs for the control of crop/stored grain pests has some difficulties (high volatility, degradability, low solubility, stability, flammability, phytotoxicity, etc.). To overcome these negative qualities, researchers have developed a formulation based on nano-emulsions for the effective control of a broad spectrum of pests of economically important crop and stored grain pests, insect vectors and pests of public importance [28,29,30].

In the present study, *C. chinensis* was comparatively more susceptible (LC_50_ = 0.94–1.41 µL/mL) to tested EOs than *C. maculatus* (LC_50_ = 1.12 to 2.02 µL/mL) within 24 to 48 h except *for M. spicata*, which showed similar toxicity (LC_50_ = 0.92 µL/mL) after 48 h. Compared to present results with those reported earlier, tested EOs showed more effectiveness with lower LC_50_ values than the EO of *Lippia alba* (Mill.) and *Callistemon lanceolatus* (Sm.) Sweet at 100 µL/L [31], *Cuminum cyminum* L. (LC_50_ = 3.50 µL/L) [32] and *Rosmarinus officinalis* (LC_50_ = 13.3 µL/L air) [32,33] against *C. chinensis*. The EOs at lower concentrations in the present study were also effective against *C. maculatus* compared to the EOs of *Artemisia herbaalba* and *Vanillosmopsis arborea* (5.2–7.7 µL/L) [34,35]. In a similar study, the EOs of *Syzygium aromaticum* and *Cinnamomum zeylanicum* (LD_50_ = 78.2 and 131 μL kg^−1^) were not superior [36] to the present study against *C. maculatus*.

The combination (1:1 ratio) of EOs was also studied for their fumigant and synergistic activities against the pulse beetle. A combination of *T. minuta* + *M. piperita* and *T. minuta* + *M. spicata* against *C. chinensis* and *C. maculatus* showed toxicity and synergistic activity. Present results confirm the findings of Erler et al. [3] who reported the binary mixtures (60–120 µL/L) of *Pimpinella anisum* L., + *Rosmarinus offcinalis* L. and *R. offcinalis* + *Thymus vulgaris* L. showed 94–100% mortality against *C. maculatus* within 48 h. Similarly, Vendan et al. [37] reported *Sitophilus oryzae* was more vulnerable to *M. piperita* along with lemon oil blend (85% mortality) with food after 72 h and followed by peppermint + orange and orange + lemon oil (61 and 53% mortality, respectively). In a similar study, a combination of EOs showed more toxicity against *Sitotroga cerealella* than individual oils alone [38]. In another study, eucalyptus oil with camphor showed promising against *Tribolium castaneum* [39]. The L-Carvone in the present study showed promising toxicity against *C. chinensis* and *C. maculatus.* However, *C. chinensis* is more susceptible to L-Carvone than *C. maculatus* but not superior to aluminum phosphide as a positive control. The present results confirm the findings of earlier studies, where carvone [40] and S-Carvone [12] exhibited better efficacy against *S. oryzae* and *C. maculatus.*

The tested EOs are dose and time-dependent, as they were significantly more repellent against the pulse beetle at higher concentrations (6–8 µL/mL) than lower (1 to 5 µL/mL). All the tested EOs showed that they were repellent against both *C. chinensis* and *C. maculatus*. The difference in the toxicity against the pulse beetle might be due to variation in the bioactive compounds and geographical distribution. In this study, *M. piperita*, *M. spicata* and *T. minuta* (8 µL/mL) showed higher repellency against *C. chinensis* and *C. maculatus*. Therefore, tested EOs in the present study at lower concentrations were superior to *Adhatoda vasica* Ness and *Chenopodium ambrosioides* L. at a higher dose (360 µL/L), which showed 100% repellency [41]. The EO of *Ocimum gratissimum* L. at 0.5–1% [42] and *Cuminum cyminum* L. seed oil at 12.5 µL/L [32] showed 73–100% repellency against *C. chinensis* after 24 h.

Present results showed that the pulse beetle is highly vulnerable to evaluated EOs in comparison to eggs. Ovicidal activity of EO against insects’ eggs varies with the sample’s concentration. Insecticidal activity of EO increases as the concentration increases and protects the grains/seeds from insect damage due to bioactive constituents and their mode of action [43]. The volatiles from the plants/EOs have reported toxicity against eggs [44,45], which may be due to infiltration of the toxic constituents/molecules through different stages of the egg and its development, which led to suffocation or a straight effect of compounds present in the volatiles. In this study, EOs at 8–12 µL/mL reported maximum ovipositional inhibition against *C. chinensis* and *C. maculatus* within 24–72 h. Therefore, these results agreed with earlier reports, where *Lippia alba* Mill. And *Callistemon lanceolatus* (Curtis) at 100 µL/L [31] reported 66.9–96%. Oviposition deterrent and *Illicium verum* and *Croton anisatum* at 17.5 µL/L [46] also showed 100% oviposition compared to the present study where the tested EO is more superior at lower concentrations. Similarly, the EO of camphor, orange, eucalyptus, mint and wintergreen (5000 µL/L) showed less activity (45–52%) against eggs of *C. maculatus* [47] as compared to the present study. In another study, higher concentrations of the EOs of *Hyptis suaveolens*, *T. minuta* (2710 µL/L) [48] and *Cymbopogon schoenanthus* (33.3 µL/L) showed 100% mortality against eggs of *C. maculatus* after 24 h [49].

Based on comparative insecticidal activities of EOs against *C. chinensis* and *C. maculatus* the result showed that the tested EOs (*M. piperita* and *T. minuta*) were more effective against *C. chinensis* (LC_50_ = 0.98–1.41 µL/mL) after 48 h of treatment as compared to *C. maculatus* (LC_50_ = 1.12–2.02 µL/mL). Similarly, L-Carvone and combinations of *T. minuta* with *M. spicata* and *M. piperita* also showed higher toxicity to *C. chinensis* as compared to *C. maculatus*. The repellence and ovipositional deterrence activity were also higher in *C. chinensis* as compared to *C. maculatus*. In this study, *C. chinensis* was more susceptible to the tested EOs than *C. maculatus*. The susceptibility/resistance may be due to the type of grains/variety and the presence of bioactive compounds in the grains. *C. chinensis* and *C. maculatus* are the major and serious pests of pulses distributed throughout the tropical and sub-tropical regions. Both the species are region-specific and do not occur simultaneously due to weather parameters. *C. chinensis* require a lower temperature (23–25 °C), and *C. maculatus* need a higher temperature (33–35 °C) for faster multiplication.

As per the reports, EOs are used in wellness, cosmetics, the food industry, etc. However, EOs can be recommended for the control of stored grain pests based on safe waiting periods and persistence studies. In this study, the main aim is to screen the EOs for their insecticidal activities (fumigant toxicity, repellent, ovipositional deterrence) against *C. chinensis* and *C. maculatus* under laboratory conditions. Further, the promising EOs need to be evaluated/validated against target pests commercially in the field (grain storage go-downs) and airtight storage bins. Based on the bio-efficacy, persistence (safe waiting periods) and residue studies, the cost-effective EO of *M. piperita* and *M. spicata* (INR 2000/ kg) can be recommended for the control of pulse beetles.

## 4. Materials and Methods

### 4.1. Essential Oils (EOs)

EOs of *Mentha piperita* L. and *Mentha spicta* L. procured from Hindustan Mint and Agro Products Pvt. Limited, Chandausi, Uttar Pradesh (INDIA) and *Tagetes minuta* L. oil from Agricultural Cooperative Society, Dharamshala, Himachal Pradesh (INDIA) extracted by steam distillation.

### 4.2. Chemical Composition of Essential Oils

#### 4.2.1. Gas Chromatography Analysis

The composition of each essential oil determined by gas chromatography (GC) on a Shimadzu GC 2010 equipped with DB-5 (J&W Scientific, Folsom, CA, USA) fused silica capillary column (30 m × 0.25 mm i.d., 0.25 µm film thickness) with a flame ionization detector (FID) [50,51]. The GC oven temperature programmed at 70 °C (initial temperature) held for 4 min and then increased at a rate of 4 °C /min to 220 °C and held for 5 min. The injector temperature was 240 °C, the detector temperature, 260 °C and the samples were injected in split mode. The carrier gas was nitrogen at a column flow rate of 1.05 mL/minute (100 kPa). The sample retention indices (RI) were determined based on homologous *n*-alkane hydrocarbons under the same conditions.

#### 4.2.2. GC-MS Analysis and Identification

The gas chromatography-mass spectrometry (GC-MS) analysis of EOs carried out using a Shimadzu QP 2010 using a DB–5 (J&W Scientific, Folsom, CA, USA) capillary column (30 m × 0.25 mm i.d., 0.25 µm film thickness). The GC oven temperature was 70 °C for 4 min and then increased to 220 °C at 4 °C /min and held for 5 min. The injector temperature was 240 °C, interface temperature was 250 °C, acquisition mass range was 800–50 amu, and the ionization energy was 70 eV. Helium is used as the carrier gas. Compounds were identified using a library search of the National Institute of Standards and Technology (NIST) database [52], as well as by comparing their RI and mass spectral fragmentation pattern with those reported in the literature [53].

### 4.3. Test Insect

Initial cultures of *C. chinensis* and *C. maculatus* obtain from the Food Protectants and Infestation Control Department, CSIR-Central Food Technological Research Institute, Mysuru, Karnataka (INDIA) for further rearing to carry out the experiments. The adults further reared in the Entomology Laboratory, CSIR-Institute of Himalayan Bioresource Technology, Palampur, Himachal Pradesh (INDIA) for more than 30 generations under controlled conditions at 25 ± 2 °C temperature, 60 ± 5% relative humidity and a photoperiod of 13:11 (L:D). The adults reared on uninfected dried mung bean/ green gram, *Vigna radiata* (L) R. Wildzek seeds in 1 L plastic jars and covered with a black muslin cloth. The adults are checked for growth regularly (15–30 day intervals) and newly emerged adults are transferred to 1 L plastic jars containing uninfested seeds for mating and egg-laying to ensure sufficient adults. The adults 1–4 days old used for bioassay and other experiments. The dead adults removed after their adult period competed, either by sieving the grains or handpicking, depending on the number of adults.

### 4.4. Fumigant Toxicity of EOs, Their Combinations/Binary Mixtures and Compounds against the Pulse Beetle

Five different test concentrations (0.8 to 4 µL/mL of air) of EOs of *T. minuta*, *M. piperita*, *M. spicata* and their combinations (1:1 ratio) were prepared for dose–response bioassay against *C. chinensis* adults to study their synergistic activity. In the case of *C. maculatus*, five different test concentrations of *T. minuta* (1–8 µL/mL), *M. piperita* (0.8 to 4 µL/mL), *M. spicata* (1–8 µL/mL) and their combinations (1–8 µL/mL) prepared for dose–response bioassay. Similarly, for the pure compound (L-Carvone), five concentrations (1 to 8 µL/mL) were prepared for dose–response bioassay against *C. chinensis* and *C. maculatus.*

The experiments were carried out on glass Desiccators (2.5 L capacity). One glass Petri dish containing five grams of insect diet (green gram) was kept at the bottom of the desiccator and released 10 adults (3–4 days old). In another Petri dish, Whatman No. 9 filter paper was placed and kept at the top of the desiccator. Five concentrations of EOs were applied on the filter paper using a micropipette, and then the lid of the desiccator was closed to make it airtight. The desiccators were kept in the controlled laboratory conditions to record the beetles’ mortality at different intervals. There were five treatments/concentrations per oil, and each treatment was replicated three times. Five concentrations of aluminum phosphide (0.5–0.9 mg/100 g grain) were tested against *C. chinensis* adults for dose–response bioassay as a positive control to compare EOs and compound. Aluminum phosphide controls stored grain pests in Food Corporation of India (FCI) godowns. Observations on mortality were recorded 24 and 48 h after treatment for EOs and their combinations/binary mixtures to calculate LC_50_ values [54] and co-toxicity coefficient [55] for binary mixtures. The co-toxicity coefficient (CTC) was calculated using the formula: CTC = [LC_50_ of A/LC_50_ of A (in a mixture)] * 100.

If the mixture gives a CTC > 100 (synergistic action), CTC < 100 (independent action) and CTC = 100 (similar action).

### 4.5. Repellent Activity of EOs against the Pulse Beetle

The repellency of EOs against *C. chinensis* and *C. maculatus* studied as per Eccles et al. [56]. Five concentrations (1–8 µL/mL, i.e., 8, 6, 4, 2, 1 µL/mL) were used and each concentration/treatment replicated five times. The Whatman No. 9 filter paper (diameter 9 cm) was cut and marked with a pencil into two halves and each labeled as treated (T) and untreated (UT). Filter papers were transferred to Petri plates (diameter 9 cm), treated with required concentrations of EOs and then allowed to air dry for 15 min. Ten adults (3–4 days old) were released in the center of the filter paper containing ten grains, and the plates were sealed with parafilm to prevent the escape of adults. The dispersal of the beetles on each side of the filter paper was recorded after 24 and 48 h of treatment.

The Percent Repellency (PR) [57] was calculated based on the formula:PR = [(Nc − Nt)/(Nc + Nt)] × 100. 
where Nc = number of insects on control half of filter paper after required exposure interval; Nt = number of insects on treated half of filter paper after required exposure interval.

The Repellent Index (RI) [58] was calculated based on RI = 2G/G + P formula. Where G = number of adults on the treated side and P = number of adults on the untreated side. The repellent index of EOs is considered as repellent, attractant or indifferent based on the mean value of RI and its respective standard deviation (SD). If the mean RI is higher than 1 + SD, the oil is an attractant, while mean RI is less than 1 − SD, the oil is repellent, and for the mean RI in between 1 − SD and 1 + SD, the oil is indifferent.

### 4.6. Ovipositional Deterrent Activity of EOs

Ovipositional deterrence of EOs against *C. chinensis* and *C. maculatus* was studied as per the method followed by Eccles et al. [56]. There are five treatments (1.25 to 20 µL/mL for *M. piperita*, 2–10 µL/mL for *M. spicata* and *T. minuta* against *C. chinensis*) and 2.5 to 30 µL/mL for all the oils against *C. maculatus*. Five different concentrations were prepared by mixing EOs in acetone for each treatment. Seeds (30 no/plate) dipped in different concentrations for 10 s, then removed and placed on filter paper to air dry for 15 min. Treated seeds were placed in a Petri plate (diameter 9 cm) and then ten adults (5 male and 5 female) of one day old were released. Petri plates were sealed with parafilm to prevent the escape of the adults. For the control, seeds were treated with acetone only. Each treatment was replicated five times. The number of eggs laid on seeds of green gram was observed from 24 to 72 h. Percent oviposition inhibition was calculated by using the formula [46].
OI = [(NC − NT)/NC] ×100 where NT = No. of eggs in untreated and NT = No. of eggs laid in treated.

### 4.7. Statistical Analysis

The data on residual toxicity bioassays of EOs, their combinations and pure compound, median lethal concentration (LC_50_), and 95% confidence limits were determined by Probit analysis [54] using SPSS software (International Business Machines Corporation, Armonk, NY, USA), version 16. Similarly, the percent repellency [57] and oviposition inhibition [46] were also calculated for different concentrations at different intervals based on the raw data and represented along with their mean and standard error. The data on percent repellency and ovipositional inhibition were subjected to one-way ANOVA by SPSS software and means were compared by Tukey’s post hoc test to know the significant differences between treatments. The normality and homogeneity of variance assumptions were tested for different parameters/variables, and no data transformations were required.

## 5. Conclusions

*M. piperita* and *M. spicata* showed promising insecticidal activities (toxicity, repellent and ovicidal) against *C. chinensis* and *C. maculatus*. Therefore, these oils can be recommended as eco-friendly and biological alternatives to synthetic pesticides to manage insect infestation in food grains stored under closed airtight conditions, particularly in bins.

## Figures and Tables

**Figure 1 molecules-27-00568-f001:**
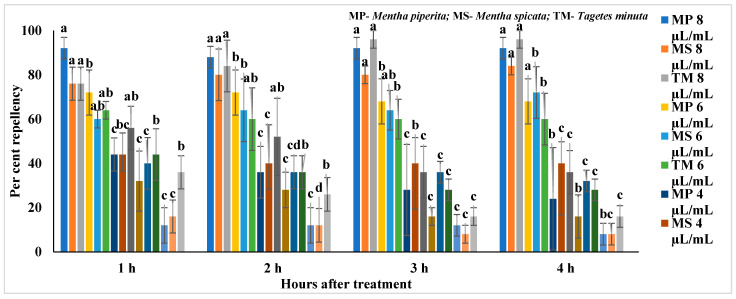
Repellence of essential oils against *Callosobruchus chinensis*; Means followed by the same letter within the error bars are not statistically different by Tukey (*p* ≤ 0.05).

**Figure 2 molecules-27-00568-f002:**
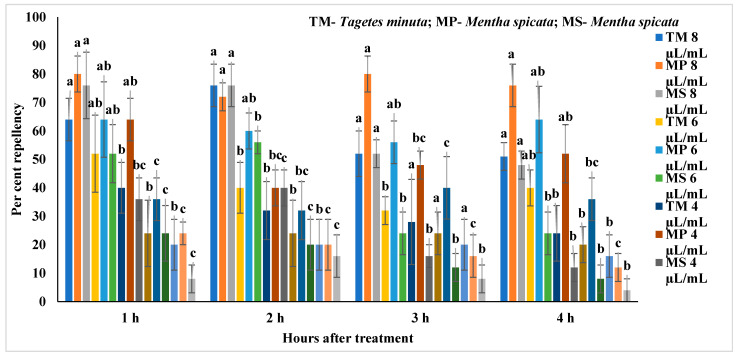
Repellence of essential oils against *Callosobruchus maculatus*; Means followed by the same letter within the error bars are not statistically different by Tukey (*p* ≤ 0.05).

**Figure 3 molecules-27-00568-f003:**
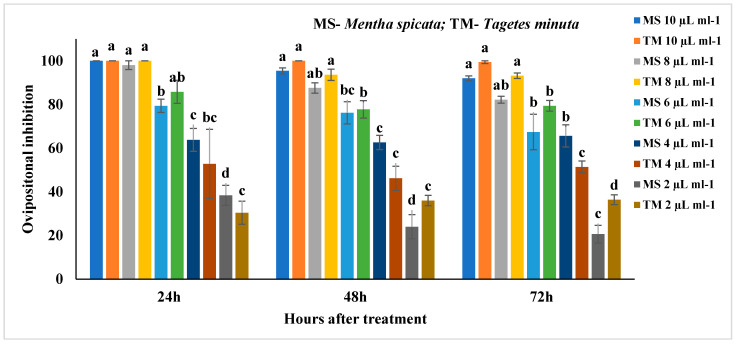
Ovipositional inhibition of *Mentha spicata and Tagetes minuta* against *Callosobruchus chinensis*; Means followed by the same letter within the error bars are not statistically different by Tukey (*p* ≤ 0.05).

**Figure 4 molecules-27-00568-f004:**
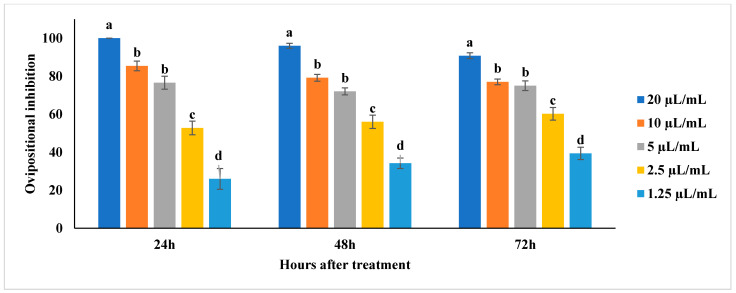
Ovipositional inhibition of *Mentha piperita* against *Callosobruchus chinensis*; Means followed by the same letter within the error bars are not statistically different by Tukey (*p* ≤ 0.05).

**Figure 5 molecules-27-00568-f005:**
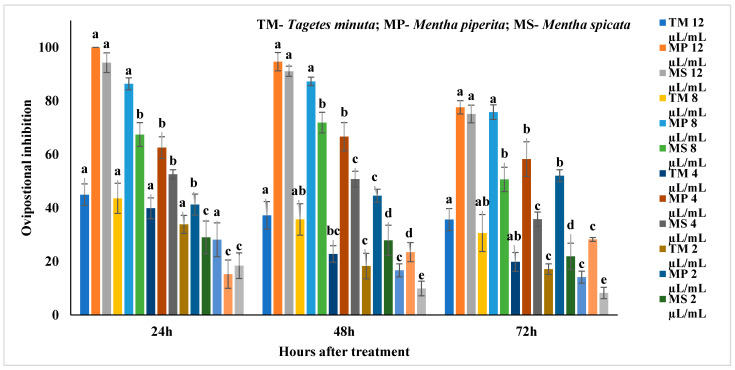
Ovipositional inhibition of essential oils against *Callosobruchus maculatus*; Means followed by the same letter within the error bars are not statistically different by Tukey (*p* ≤ 0.05).

**Table 1 molecules-27-00568-t001:** Fumigant toxicity of essential oils and their combinations against *Callosobruchus chinensis*.

Time	LC_50_ (µL/mL)	CL (µL/mL)	Slope ± SE	Chi- Square	*p*-Value	Co-Toxicity Coefficient	Interaction Type
*Tagetes minuta*							
24 h	3.49	2.77–5.36	2.28 ± 0.53	0.21	0.98	-	-
48 h	1.41	1.09–1.71	2.90 ± 0.49	0.87	0.83	-	-
*Mentha piperita*							
24 h	2.06	2.46–3.47	3.44 ± 0.62	2.04	0.56	-	-
48 h	0.98	0.66–1.23	2.82 ± 0.52	3.41	0.33	-	-
*Mentha spicata*							
24 h	1.88	1.40–2.42	2.00 ± 0.45	5.10	0.16	-	-
48 h	0.94	0.62–1.20	2.74 ± 0.52	2.62	0.45	-	-
*Tagetes minuta* + *Mentha piperita*							
24 h	1.35	1.02–1.64	2.86 ± 0.49	0.89	0.83	258.52	Synergistic
48 h	0.87	0.58–1.10	3.12 ± 0.58	1.54	0.67	162.07	Synergistic
*Tagetes minuta* + *Mentha spicata*							
24 h	1.53	1.29–1.76	4.22 ± 0.59	4.60	0.20	228.10	Synergistic
48 h	1.07	0.86–1.25	4.48 ± 0.68	1.23	0.75	131.78	Synergistic

CL: Confidence limits.

**Table 2 molecules-27-00568-t002:** Fumigant toxicity of essential oils and their combination against *Callosobruchus maculatus*.

Time	LC_50_ (µL/mL)	CL (µL/mL)	Slope ± SE	Chi- Square	*p*-Value	Co-Toxicity Coefficient	Interaction Type
*Tagetes minuta*							
24 h	3.42	2.85–4.06	3.25 ± 0.44	5.1	0.16	-	-
48 h	2.02	1.63–2.43	3.20 ± 0.44	4.3	0.23	-	-
*Mentha piperita*							
24 h	1.76	1.50–2.02	4.03 ± 0.57	4.80	0.19	-	-
48 h	1.12	0.86–1.35	3.36 ± 0.55	2.78	0.43	-	-
*Mentha spicata*							
24 h	2.74	1.69–3.97	1.33 ± 0.33	1.35	0.72	-	-
48 h	0.92	0.47–1.30	2.22 ± 0.46	1.17	0.76	-	-
*Tagetes minuta* + *Mentha piperita*							
24 h	4.93	4.11–6.06	3.04 ± 0.49	2.59	0.46	69.37	Independent
48 h	2.40	2.03–2.80	4.30 ± 0.55	3.20	0.36	84.17	Independent
*Tagetes minuta* + *Mentha spicata*							
24 h	2.42	1.91–2.94	2.77 ± 0.40	5.31	0.15	141.32	Synergistic
48 h	1.42	1.05–1.77	3.01 ± 0.48	1.71	0.64	142.25	Synergistic

CL: Confidence limits.

**Table 3 molecules-27-00568-t003:** Fumigant toxicity of L-Carvone against *Callosobruchus chinensis* and *C. maculatus*.

	*C. chinensis*				
Time	LC_50_ (µL/mL)	CL (µL/mL)	Slope ± SE	Chi-square	*p*-value
24 h	3.61	2.65–4.99	1.69 ± 0.35	0.86	0.83
26 h	2.92	2.69–3.92	1.72 ± 0.34	0.35	0.95
28 h	2.16	1.41–2.90	1.68 ± 0.34	2.06	0.56
36 h	1.33	0.82–1.79	2.08 ± 0.39	3.45	0.33
48 h	1.19	0.77–1.57	2.46 ± 0.44	1.60	0.66
	*C. maculatus*				
24 h	6.72	5.40–9.41	2.62 ± 0.50	3.91	0.27
26 h	5.90	4.87–7.64	2.88 ± 0.50	3.03	0.39
28 h	5.39	4.50–6.69	3.10 ± 0.50	2.90	0.40
36 h	3.76	3.14–4.49	3.20 ± 0.45	4.70	0.19
48 h	3.56	3.00–4.20	3.46 ± 0.47	5.10	0.16

CL: Confidence limits.

**Table 4 molecules-27-00568-t004:** Repellent index of essential oils against *Callosobruchus chinensis*.

EOs	Conc. (µL/mL)	Repellent Index (RI); (Hours after Treatment (* Mean ± SD))			
		1 h	2 h	3 h	4 h
*Tagetes* *minuta*	8	0.24 ± 0.17 (R)	0.16 ± 0.26 (R)	0.04 ± 0.09 (R)	0.04 ± 0.09 (R)
	6	0.36 ± 0.09 (R)	0.40 ± 0.32 (R)	0.40 ± 0.20 (R)	0.44 ± 0.26 (R)
	4	0.44 ± 0.22 (R)	0.48 ± 0.39 (R)	0.64 ± 0.26 (R)	0.76 ± 0.22 (R)
	2	0.56 ± 0.26 (R)	0.64 ± 0.17 (R)	0.72 ± 0.11 (R)	0.88 ± 0.11 (R)
	1	0.64 ± 0.17 (R)	0.73 ± 0.17 (R)	0.84 ± 0.09 (R)	0.92 ± 0.11 (I)
*Mentha piperita*	8	0.08 ± 0.11 (R)	0.12 ± 0.11 (R)	0.08 ± 0.11 (R)	0.08 ± 0.11 (R)
	6	0.28 ± 0.23 (R)	0.28 ± 0.23 (R)	0.32 ± 0.23 (R)	0.32 ± 0.23 (R)
	4	0.64 ± 0.26 (R)	0.64 ± 0.26 (R)	0.72 ± 0.46 (R)	0.76 ± 0.52 (I)
	2	0.68 ± 0.30 (R)	0.72 ± 0.18 (R)	0.84 ± 0.09 (R)	0.84 ± 0.22 (I)
	1	0.88 ± 0.18 (I)	0.88 ± 0.18 (I)	0.88 ± 0.11 (R)	0.92 ± 0.11 (I)
*Mentha spicata*	8	0.24 ± 0.17 (R)	0.20 ± 0.20 (R)	0.20 ± 0.20 (R)	0.16 ± 0.22 (R)
	6	0.40 ± 0.14 (R)	0.36 ± 0.09 (R)	0.36 ± 0.09 (R)	0.28 ± 0.11 (R)
	4	0.56 ± 0.22 (R)	0.60 ± 0.14 (R)	0.60 ± 0.14 (R)	0.60 ± 0.32 (R)
	2	0.60 ± 0.24 (R)	0.64 ± 0.17 (R)	0.64 ± 0.22 (R)	0.68 ± 0.27 (R)
	1	0.84 ± 0.09 (R)	0.88 ± 0.11 (R)	0.92 ± 0.11 (I)	0.92 ± 0.11 (I)

* Mean of five replications; R—Repellent (RI less than 1 − SD), I—Indifferent (RI in between 1 − SD and 1 + SD), A—Attractant (RI greater than 1 + SD).

**Table 5 molecules-27-00568-t005:** Repellent index of essential oils against *Callosobruchus maculatus*.

EOs	Conc. (µL/mL)	Repellent Index (RI) (Hours after Treatment (* Mean ± SD))			
		1 h	2 h	3 h	4 h
*Tagetes minuta*	8	0.36 ± 0.17 (R)	0.24 ± 0.17 (R)	0.48 ± 0.18 (R)	0.49 ± 0.10 (R)
	6	0.48 ± 0.30 (R)	0.60 ± 0.20 (R)	0.68 ± 0.11 (R)	0.60 ± 0.14 (R)
	4	0.60 ± 0.20 (R)	0.68 ± 0.23 (R)	0.72 ± 0.33 (I)	0.76 ± 0.22 (R)
	2	0.76 ± 0.26 (I)	0.76 ± 0.26 (I)	0.76 ± 0.17 (R)	0.80 ± 0.14 (R)
	1	0.80 ± 0.20 (I)	0.80 ± 0.20 (I)	0.80 ± 0.20 (I)	0.84 ± 0.17 (I)
*Mentha piperita*	8	0.20 ± 0.14 (R)	0.28 ± 0.11 (R)	0.20 ± 0.14 (R)	0.24 ± 0.17 (R)
	6	0.36 ± 0.30 (R)	0.40 ± 0.14 (R)	0.44 ± 0.17 (R)	0.36 ± 0.26 (R)
	4	0.36 ± 0.17 (R)	0.60 ± 0.14 (R)	0.52 ± 0.11 (R)	0.48 ± 0.23 (R)
	2	0.64 ± 0.17 (R)	0.68 ± 0.23 (R)	0.60 ± 0.24 (R)	0.64 ± 0.17 (R)
	1	0.88 ± 0.23 (I)	0.84 ± 0.17 (I)	0.84 ± 0.17 (I)	1.00 ± 0.20 (I)
*Mentha spicata*	8	0.24 ± 0.26 (R)	0.24 ± 0.17 (R)	0.48 ± 0.11 (R)	0.52 ± 0.11 (R)
	6	0.48 ± 0.23 (R)	0.44 ± 0.09 (R)	0.76 ± 0.17 (R)	0.76 ± 0.17 (R)
	4	0.64 ± 0.17 (R)	0.60 ± 0.14 (R)	0.84 ± 0.09 (R)	0.88 ± 0.11 (R)
	2	0.76 ± 0.22 (R)	0.80 ± 0.20 (I)	0.88 ± 0.11 (R)	0.92 ± 0.11 (I)
	1	0.92 ± 0.11 (I)	0.84 ± 0.17 (I)	0.92 ± 0.11 (I)	0.96 ± 0.09 (I)

* Mean of five replications; R—Repellent (RI less than 1 − SD), I—Indifferent (RI in between 1 − SD and 1 + SD), A—Attractant (RI greater than 1 + SD).

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
