# Peer review of "Chemical Composition and Insecticidal Activities of Essential Oils against the Pulse Beetle"

_molecules, 2022, doi:10.3390/molecules27020568_

Round 1

Reviewer 1 Report

Dear Authors,

I have gone through your research article, Chemical Composition and Insecticidal Activities of Essential Oils against Pulse Beetle submitted to Molecules. No doubt, it's a nice piece of work. But before final publication, I strongly recommend a regorous improvement of this manuscript. Please find my general & specific comments below:

General Comments:

[1] Lots of typos & Grammatical errors & lack of consistency throughout the manuscript.

[2] Extensive english corrections are needed by a native english expert.

Specific Comments:

[1] P 1, L 30: ''........... damage to cowpea, gram, soybean and pulses in storage...... ''. What do you mean by ''and pulses''? because cowpea, gram and soybean also in the group of pulses. Please revise the sentence.

[2] P 1, L 37: ''Thepulse beetle'' should be ''The pulse beetle''

[3] P2, L45-47: Very difficult to understand the fact. Please revise the sentence to make it clear.

[4] P2 L52: ''good potential" better to write ''potential''

[5] P2, L55-56: So direct & also the statement not true actually (please search in google and will find lots of example). Please revise the statement.

[6] P2, L57-61: Very poor sentence construction and doesn't make any sense. Please the text & make it more simple.

[7] P2, L83; P3, L99; P4 L123, P6 L157, L182; P7 L199: All Scientific name should be Italic. Please see other sections too.

[8] P2, L86-88: .......24h and 48 h......not clear. Is it after application or before application. Please revise the sentence. 

[9] P4 L117: ''i.e.'' should be italic

[10] P4 L124: at1 should be at 1.

[11] P7 L192: ''.................inhibiting 100% at 20 μL ml^−1 after 24 h.......... Where you find this concentration? I didn't see it in your figure 4.

[12] P7 L201, L213: Figure 5 is missing. Please add Figure 5 or delete the section.

[13] Discussion: Please be consistent in writing units throughout the manuscript. e.g. µL ml^-1 Or µL mL^-1 Or uL mL^-1. Which one is correct?

[14] P9, L295: at12.5 should be at 12.5

[15] P9, L298-300: Doesn't make any sense. Please revise.

[16] P9, L310: "L. alba and C. lanceolatus ''. Better to write the full name of the genus if you write it for the first time.

[17] P10, L330-331, L339-340: Degree symbol should not be superscript. 

[18] P12, L433-435: Doesn't make any sense. Please revise.

Reviewer 2 Report

The topic has great merit despite its lack of a robust discussion of the results. Authors should try to explain better the differential efficacy of the treatments on the two bruchid species. Why did this happen and what is the ecological importance? These species do not always occur together, in fact they mostly occur geographically separated.  Also, even if essential oils are effective, they are also very caustic and could be unsafe to users and consumers. These are issues that need to be brought ino the discussion. What is the intended point or sale of application? Is it practical?

The grammar MUST be vastly improved if it is going to be accepted for publication. It does not matter who handles this, authors or editors, it must be done. 

Round 2

Reviewer 2 Report

I had a problem reviewing large chunks of tracked changes. The norm is to provide a numbered listing (with the pages indicated) of the changes made, or why not. This sort of revised manuscript with red all over the place is unacceptable. 
I can see the effort that was put into the revision, and that included new errors and non-addressed queries from my initial review.

There is a clear difficulty trying to address the two bruchids in one paper. That clearly is difficult for the authors who make recommendations and other (highlighted) statements that have not been proven in the study or supported by relevant citations.

The English grammar remains a difficult area that must be addressed if this is going to ever be published.

Round 3

Reviewer 2 Report

I can tell the authors have made a concerted effort to address the major points I raised. The paper can now be accepted with minor editorial work that may be needed. The paper is clearly not perfect, but we have tried to get it to as close as possible to that as possible. Good job!